# Estimation of Genetic Variances and Stability Components of Yield-Related Traits of Green Super Rice at Multi-Environmental Conditions in Pakistan

Imdad Ullah Zaid [†], Nageen Zahra [†], Madiha Habib [iD], Muhammad Kashif Naeem, Umair Asghar, Muhammad Uzair [iD], Anila Latif, Anum Rehman, Ghulam Muhammad Ali and Muhammad Ramzan Khan *

National Institute for Genomics and Advanced Biotechnology (NIGAB), National Agricultural Research Centre, Islamabad 45500, Pakistan; imdadcas@gmail.com (I.U.Z.); nageenzahra@hotmail.com (N.Z.); madihahabib217@gmail.com (M.H.); kashifuaar@gmail.com (M.K.N.); umairasghar308@gmail.com (U.A.); uzairbreeder@gmail.com (M.U.); anilalatif87@gmail.com (A.L.); anumrehman92@hotmail.com (A.R.); chairman@parc.gov.pk (G.M.A.)
* Correspondence: mrkhan@parc.gov.pk; Tel.: +92-5190733808
† These authors contributed equally to this work.

**Abstract:** Identifying adopted Green Super Rice (GSR) under different agro-ecological locations in Pakistan is crucial to sustaining the high productivity of rice. For this purpose, the multi-location trials of GSR were conducted to evaluate the magnitude of genetic variability, heritability, and stability in eight different locations in Pakistan. The experimental trial was laid out in a randomized complete block (RCB) design with three replications at each location. The combined analysis of variance (ANOVA) manifested significant variations for tested genotypes (g), locations (L), years (Y), genotype × year (GY), and genotype × location (GL) interactions revealing the influence of environmental factors (L and Y) on yield traits. High broad-sense heritability estimates were observed for all the studied traits representing low environmental influence over the expression of traits. Noticeably, GSR 48 showed maximum stability than all other lines in the univariate model across the two years for grain yield and related traits data. Multivariate stability analysis characterized GSR 305 and GSR 252 as the highest yielding with optimum stability across the eight tested locations. Overall, Narowal, Muzaffargarh, and Swat were the most stable locations for GSR cultivation in Pakistan. In conclusion, this study revealed that G×E interactions were an important source of rice yield variation, and its AMMI and biplots analysis are efficient tools for visualizing the response of genotypes to different locations.

**Keywords:** variability; heritability; univariate stability analysis; AMMI; GGE biplot analysis

## 1. Introduction

Rice is considered an important staple food crop across the globe, including in Pakistan. In 2020, Pakistan produced ten percent of the world's rice and ranked among the top ten rice producers worldwide (FAO, 2020) [1]. Rice is the sixth-largest export commodity of Pakistan. Pakistan exports more than 4.59 million t (making up 8% of the world's total rice trade), equivalent to 2.3 billion USD, which accounted for 10% of the world's total exports, ranking Pakistan third-largest rice exporter in terms of volume and value (International Trading Center (ITC) 2020). Rice cultivars grown in Pakistan are mainly divided into IIRI type, Basmati, and non-Basmati type. Basmati rice, an exclusive trademark of Pakistan with elongated and slender grains, soft and fluffy texture when cooked, and an aromatic taste, is one of the most appealing high-end rice in the international market. From September to December 2020, Basmati rice increased its footprint in the European market, retaining the minimum level of pesticide contamination per the European Union's standard. Moreover, rice exports rose in the country during November 2020, with 78,160 t valuing USD 76 m

from 43,032 t to fetching USD 41 m in October 2020. According to the Rice Exporters Association of Pakistan (REAP), exports of coarse rice also expanded sharply to 379,944 t, with earnings of USD 154 m in November compared to 220,674 t fetching USD 98 m in October 2020.

Unfortunately, in Pakistan, the unavailability of certified seeds, diseases, and insect pests attack, uneven and limited distribution of water for paddy irrigation, fertilizer management, and post-harvest losses are critical factors in rice production. Moreover, occurrences of floods, temperature rises, droughts, and unusual rainfalls subsequently increase the skirmishes between rice production and environmental resources. Under these consequences, the fundamental breeding objective is to develop rice cultivars that reveal green traits, i.e., tolerance against multiple stresses, high nutrients-yield potential, and fertilizer–water-use efficiencies.

In the term "Green Super Rice (GSR)", the word "Green" means environmentally friendly as it grows more grains under fewer inputs while "Super" means more stress-tolerant. In the light of growing fluctuating resources, the development and adaptation of GSR also represent resources-saving, high-yielding, efficient, and ecologically stable rice [2]. Recently, 552 GSR advanced lines were introduced at National Institute for Genomics and Advanced Biotechnology (NIGAB) National Agriculture Research Council (NARC), Islamabad (Pakistan), to develop rice cultivars that retain sustainable yield even under unfavorable environmental conditions.

Before releasing a new variety for commercial purposes, plant breeders usually evaluate the set of genotypes across multi-environments. A stable genotype produces the expected yield in a particular environment [3]. The stronger a genotype–environment interaction is, the more unpredictable it is to assess the performance of a genotype in multi-environments [4]. Selection of a particular genotype becomes difficult due to genotype × environment interaction [5]. Hence, it is significant to assess the adaptation and stability of a group of genotypes before commercial release. Various statistical methods that have been developed for this purpose are divided into parametric and non-parametric stability statistics. Parametric stability statistics is further divided into univariate and multivariate methods. The univariate methods include Wricke's ecovalence ($W_i^2$) [3], Shukla's stability variance ($\sigma^2$) [6], coefficient of variance ($CV$) [7], Environmental variance ($S^2$) [8], Mean-variance component ($\theta$) [9], GE variance component ($\theta'$) [10], Regression coefficient ($b_i$) [11], and many others. The multivariate methods imply the additive main effects and multiplicative interaction (AMMI) model [12] and the GGE biplot method [13]. Multivariate methods can effectively predict the genotype × environment interactions by using the approaches such as the 'which-won-where' pattern, identifying mega environments, ideal genotypes across different testing environments, and ranking environments [14]. Non-parametric methods include Nassa and Huhn's and Huhn's statistics (S) [15], Kang's rank-sum (KR) [16], TOP-Fox (TOP) [17], Thennarasu's non-parametric statistics (NP) [18], and Genotype stability index (GSI) [19].

The present study aims to identify superior rice genotypes with stable yield performance over eight different locations for two consecutive years by evaluating the efficacy of various univariate and multivariate stability parameters.

## 2. Materials and Methods

### 2.1. Plant Material

Five experimental genotypes and two commercial check cultivars were evaluated at eight different locations using RCBD with three replications in three provinces of Pakistan (Table 1). The experimental rice genotypes were: GSR-48, GSR-82, GSR-112, GSR-252, and GSR-305. The check cultivars evaluated were IRRI-6 and Kissan Basmati. The GSR lines were selected based on the two-year agro-morphological performance for yield and yield-related traits at the National Institute for Genomics and Advanced Biotechnology (NIGAB) National Agriculture Research Council (NARC) Islamabad, Pakistan.

**Table 1.** Code for genotypes name and locations evaluated during the two years.

| Genotypes with Codes | Locations with Environment Codes | Years |
|---|---|---|
| GSR-48 = G1 | Pindi Bhattian = E1 | |
| GSR-82 = G2 | Kala Shah Kaku = E2 | |
| GSR-112 = G3 | Narowal = E3 | |
| GSR-252 = G4 | Swat = E4 | |
| GSR-305 = G5 | Islamabad = E5 | 2020, 2021 |
| IRRI6 = G6 | Dera Ismail Khan = E6 | |
| Kissan Basmati = G7 | Muzaffargarh = E7 | |
| | Dokri = E8 | |

### 2.2. Experimental Location

All genotypes were evaluated at eight different locations: Soil Salinity Research Institute Pindi Bhattian; Rice Research Institute Kala Shah Kaku, Narowal, and Muzaffargarh in Punjab; Agriculture Research Institute Swat and Dera Ismail Khan in Khyber Pakhtunkhwa; Dokri in Sindh; and National Agriculture Research Centre (NARC) Islamabad for two years cropping season of 2020–2021. Climatic characteristics (average rainfall and temperature) of test locations for 2020–2021 from transplantation to harvesting are given in Table 2.

### 2.3. Experimental Procedures and Cultural Practices

In the first week of each year June 2020 and 2021, 1000 cleaned seeds of GSR lines with two check cultivars were sown on nursery trays with 98 holes, where each hole was seeded with two healthy seeds. The plastic trays were filled with a mix of 70% sandy clay loam soil and 25% peat moss. The trays were labeled with genotype code and name, respectively. The 30-day-old rice seedlings were shifted to paddy fields at eight different locations and transplanted manually. Transplantations of all rice genotypes were performed on the third of July (2020 and 2021) in a straight-rows method in three replications at each location. Each plot was set with a net size of 2.1 m × 0.90 m containing five rows with eight seedlings per row. There was a 17 cm row-to-row and 20 cm plant-to-plant spacing within the plot. All the yield and yield-related traits were measured at the physiological maturity stage. Data were collected from five randomly selected plants from each plot in each replication. The plant height (PH) of each genotype was measured with the help of a meter rod in centimeters (cm). Tillers per plant (TPP) was determined by counting all productive tillers' numbers. Straw yield per plant (SYPP) and grain yield per plant (GYPP) were recorded with 14% moisture content. Nitrogenous fertilizers were applied in three splits (after seven days, 37 days, and 60 days of transplantation); phosphorus and potash were used in full doses after the two weeks of transplantation. During the rice growth stages, weeds were removed by two times hand-weeding. However, neither herbicides nor insecticides were applied in the experimental trials.

### 2.4. Statistical Analysis

2.4.1. Analysis of Variance

The obtained morphological data of five GSR lines along with check cultivars at eight different locations over two years were subjected to the combined ANOVA, using R statistical software version 4.1.1. Furthermore, ANOVA results were used to determine the effect of genotypes (G), locations (L), replications (R), and years (Y) effect and the magnitude of the G × L, G × Y, and G × L × Y interactions.

**Table 2.** Mean temperature, rainfall, and soil texture of each experimental site during 2020 and 2021.

| Locations | Month | July | | August | | September | | October | | November | | Soil Type |
|---|---|---|---|---|---|---|---|---|---|---|---|---|
| | Year | Temp (°C) | Rain (mm) | Temp (°C) | Rain (mm) | Temp (°C) | Rain (mm) | Temp (°C) | Rain (mm) | Temp (°C) | Rain (mm) | |
| NARC | 2020 | 35 | 162.5 | 32 | 165.1 | 31 | 68.5 | 29 | 22.8 | 22 | 12.7 | loam |
| | 2021 | 28 | 174 | 26 | 162 | 25 | 73 | 21 | 31 | 15 | 39 | |
| Swat | 2020 | 33 | 89.8 | 35 | 55.8 | 28 | 91.8 | 23 | 36.8 | 19 | 64.4 | sandy |
| | 2021 | 28 | 55.8 | 27 | 55.8 | 25 | 27.9 | 19 | 20.3 | 13 | 15.2 | |
| Kala Shah Kaku | 2020 | 36 | 50.6 | 33 | 57.2 | 32 | 39.2 | 32 | 3.2 | 23.5 | 2.8 | silty clay |
| | 2021 | 31 | 134.6 | 31 | 124.4 | 30 | 55.8 | 25 | 12.7 | 20 | 5 | |
| Pindi Bhattian | 2020 | 34 | 71.1 | 33 | 67.5 | 32 | 45.3 | 32 | 6.4 | 24 | 4.4 | sandy loam |
| | 2021 | 35 | 19.2 | 36 | 91.4 | 34 | 40.6 | 30 | 7.6 | 24 | 5 | |
| Narowal | 2020 | 36 | 160.5 | 33 | 232.6 | 35 | 139.9 | 32 | 14.3 | 23.5 | 12.4 | silty, loamy |
| | 2021 | 33 | 89 | 31 | 59 | 28.5 | 43 | 23 | 11 | 19 | 4 | |
| Muzaffargarh | 2020 | 40 | 37.5 | 37 | 43.7 | 36 | 20.8 | 36 | 2.21 | 28.5 | 1.8 | salinity |
| | 2021 | 39.2 | 52 | 38.1 | 40 | 37.2 | 19 | 34.4 | 2 | 28.3 | 3 | |
| Dokri | 2020 | 41 | 118.8 | 38.5 | 106.8 | 37 | 50.1 | 33 | 12.23 | 26.5 | 5.8 | sandy clay loam |
| | 2021 | 41.1 | 41 | 39 | 24 | 38.3 | 9 | 35.7 | 2 | 30.1 | 2 | |
| Dera Ismail Khan | 2020 | 33 | 61 | 32 | 58 | 30 | 18 | 25 | 5 | 19 | 2 | sandy/loamy sand |
| | 2021 | 41.4 | 60 | 38.4 | 57 | 37.3 | 18 | 36.6 | 5 | 33.2 | 2 | |

### 2.4.2. Genetic Parameters

Genetic and environmental effects among the genotypes for traits were measured by using their mean sum of squares [20]. The heritability estimate was categorized as low (0–30%), medium (30–60%), and high (>60%).

(a) Genotypic variance

$$\sigma^2 g = \frac{\text{GMS} - \text{EMS}}{r}$$

Here, GMS is the genotype mean square and EMS denotes the error mean square, and r is the number of replications of genotypes.

(b) Phenotypic variance

$$\sigma^2 p = \sigma^2 g + \sigma^2 e$$

Here, $\sigma^2 p$ is the phenotypic variance, $\sigma^2 g$ is the genotypic variance, and $\sigma^2 e$ is the environmental variance.

(c) Environmental variance

$$\sigma^2 e = \frac{\text{EMS}}{r}$$

Here, $\sigma^2 e$ is the environmental variance, EMS is the error mean square, and r is the number of replications of genotypes.

(d) $H^2$

$$h_B^2 = \frac{\sigma^2 g}{\sigma^2 p}$$

where $h_B^2$ is the broad-sense heritability, which is equal to the ratio of $\sigma^2 g$ (genotypic variance) and $\sigma^2 p$ (phenotypic variance).

### 2.4.3. Estimation of Stability Parameters

The univariate and multivariate parametric stability analyses were performed to assess genotype yield and yield-related traits across multiple environments and predict stable genotypes. Both univariate and multivariate stability analyses were performed year-wise due to the presence of significant variation between the year effect.

### 2.4.4. Univariate Stability Analysis

Univariate stability of the 7 genotypes for plant height, number of tillers per plant, grain yield per plant, and straw yield per plant was calculated by using AMMI Stability Value (ASV) [21] and AMMI Stability Index (ASI) [22], Shukla's stability variance ($\sigma^2$) [6] and Wricke's ecovalence ($Wi^2$) [3].

1. AMMI Stability Value (ASV)

As suggested by Purchase et al. [21], AMMI Stability Value (ASV) parameter for stability assessment is calculated by the following equation

$$\text{ASV} = \sqrt{\left( \frac{\text{SS}_{\text{IPCA1}}}{\text{SS}_{\text{IPCA2}}} \ (\text{IPCA1}) \right)^2 + (\text{IPCA2})^2}$$

$\text{SS}_{\text{IPCA1}}$ and $\text{SS}_{\text{IPCA2}}$ are the sum of squares in the first two principal component interactions. IPCA1 and IPCA2 are the scores of genotypes in the first and second principal components interactions.

2. AMMI Stability Index (ASI)

Jambhulkar et al. [22] suggested the AMMI-model based AMMI Stability Index (ASI), which is calculated by using the following equation:

$$\text{ASI} = \sqrt{\left[ (\text{IPCA1} \times \theta_1^2)^2 + (\text{IPCA2} \times \theta_2^2)^2 \right]}$$

IPCA1 and IPCA2 are the values of the first two principal component interactions and $\theta_1^2$ and $\theta_2^2$ are the values of the percentage sum of square explained by these two components.

3. Wricke's Ecovalence

Wricke [3] introduced the idea of ecovalence parameter to calculate the share of each genotype to the sum of squares of genotype × environment interaction by using the following equation:

$$W_i^2 = \sum \left(X_{ij} - \overline{X}_{i.} - \overline{X}_{.j} + \overline{X}_{..}\right)^2$$

Here, $X_{ij}$ represents the mean of $i$th genotype in the $j$th environment, $\overline{X}_{i.}$ is the mean of the yield of $i$th genotype, $\overline{X}_{.j}$ is the mean of the yield of the $j$th environment, and $\overline{X}_{..}$ is the grand mean.

4. Shukla's Stability Variance

Shukla [6] proposed the Shukla's stability variance of genotypes across different environments based on the following equation:

$$\sigma^2 = \left[\frac{p}{(p-2)(q-1)}\right] W_i^2 - \frac{\sum W_i^2}{(p-1)(p-2)(q-1)}$$

Here, p and q represent the genotypes and environments number while $W_i^2$ is the Wricke's ecovalence of the $i$th genotype.

### 2.4.5. Multivariate Stability Analysis

Multivariate stability analysis; AMMI [23] and GGE biplot [13] were performed to identify the ideal genotype across each testing environment with high performance and stability, mega-environments, and understanding of the genotype × environment interactions.

### 2.4.6. Additive Main Effect and Multiplicative Interaction (AMMI) Model

In the present study, multivariate stability based on the AMMI model was assessed for G×E interaction and stability analysis to predict the stability of GSR lines. The AMMI model combines the application of pooled ANOVA to evaluate the additive main effects; then factorization of a complex matrix (SVD) is applied to the total error for computing interaction principal components (IPCs). We estimated the additive main effect and AMMI model in R using the metan library [24]. As suggested by Zobel et al. [23] the base of the additive main effect and multiplicative interaction (AMMI) model was computed as follows:

$$Y_{ij} = \mu + \alpha_i + \beta_j + \sum_{k=1}^{n} \lambda_k \gamma_{ik} \delta_{jk} + \varepsilon_{ij}$$

where $Y_{ij}$ is the mean performance of $i$th genotype in the $j$th individual environment, $\mu$ is the overall mean, $\alpha_i$ is the *fixed effect of the GSR line*, $\beta_j$ is the *environmental effect*, $n$ is the number of IPCA kept in the AMMI model, $\lambda_k$ is the singular value for IPC axis $k$, $\gamma_{ik}$ is the $i$th genotype eigenvector value for IPC axis $k$, $\delta_{jk}$ is the $j$th environment eigenvector value for IPC axis $k$, and $\varepsilon_{ij}$ is the average residual.

### 2.4.7. Biplot Analysis

After ranking the most adoptable GSR lines with the AMMI model, a study of the sustainable phenotypic reliability of the multi-locations analysis of the biplot graphic was designed. Biplots are performance and stability-related graphs where factors of both genotypes and locations are plotted on the same axis so the inter-relationships can be visualized.

In our constructed biplots, the abscissa represents the variables that affect the values of a genotype, and the ordinate is the first interaction axis (IPCA1). The GSR line with IPCA1 close to zero will be considered a stable and "ideal" GSR line while low stability will be associated with low productivity [25,26].

## 3. Results

### 3.1. Combined Analysis of Variance

Table 3 and Supplementary Table S1 represent the combined ANOVA of rice genotypes for plant height, tillers per plant, straw yield per plant, and grain yield per plant across eight different locations (Supplementary Table S1). The mean square of genotypes showed significant differences ($p \leq 0.01$) for traits. The mean square of locations, years, and genotypes by locations, genotypes by years, and genotypes by locations by years (G × L interaction, G × Y interaction, and G × L × Y interaction) showed significant differences ($p \leq 0.01$) for traits. In our study, the significant G × L interaction effects revealed that rice genotypes responded differently against fluctuation in the environment, which indicated the necessity of testing rice genotypes at multiple locations. Moreover, interaction among genotypes, locations, and years also revealed a highly significant ($p \leq 0.01$) difference for studied traits. Therefore, further general adaptability and stability analysis across genotypes should be followed before their selection.

**Table 3.** Estimation of significant levels for yield and related traits of seven rice genotypes revealed by combined ANOVA.

| Source of Variation | df | Plant Height | Tillers Per Plant | Grain Yield per plant | Straw Yield per plant |
|---|---|---|---|---|---|
| Genotype | 6 | $1.07 \times 10^{-52}$ | $7.07 \times 10^{-4}$ | $3.14 \times 10^{-16}$ | $1.54 \times 10^{-15}$ |
| location | 7 | $2.76 \times 10^{-12}$ | $2.12 \times 10^{-52}$ | $1.05 \times 10^{-86}$ | $2.74 \times 10^{-60}$ |
| Year | 1 | $1.78 \times 10^{-3}$ | $2.43 \times 10^{-2}$ | $1.52 \times 10^{-72}$ | $7.03 \times 10^{-36}$ |
| Replication | 2 | $7.59 \times 10^{-6}$ | $6.57 \times 10^{-1}$ | $2.30 \times 10^{-1}$ | $9.54 \times 10^{-3}$ |
| Genotype: Location | 42 | $3.78 \times 10^{-5}$ | $4.77 \times 10^{-3}$ | $1.97 \times 10^{-10}$ | $3.23 \times 10^{-7}$ |
| Genotype: Year | 6 | $1.76 \times 10^{-6}$ | $6.72 \times 10^{-2}$ | $1.88 \times 10^{-3}$ | $6.03 \times 10^{-3}$ |
| Genotype:Location:Year | 42 | $2.84 \times 10^{-7}$ | $2.22 \times 10^{-3}$ | $4.11 \times 10^{-6}$ | $3.97 \times 10^{-15}$ |

df = degree of freedom.

### 3.2. Analysis of Genetic and Phenotypic Variances

In our study, the phenotypic variance for plant height, tillers per plant, grain yield per plant, and straw yield per plant were distributed into genotypic and environmental variances (Table 4). A low magnitude of genotypic coefficient of variation was found in the corresponding phenotypic coefficient of variation for all traits studied. The broad-sense heritability for four yield traits ranged from 75.3% (tillers per plant) to 98.7% (plant height), respectively. Accordingly, all the yield-related traits considered in our study showed high heritability (>60%), constituting a high breeding value with more additive genetic effects, which is important for rice grain yield enhancement.

**Table 4.** Estimation of genetic parameters in rice genotypes for yield and yield-related traits.

| Genetics Parameters | Plant Height | Tillers Per Plant | Grain Yield Per Plant | Straw Yield per Plant |
|---|---|---|---|---|
| $Vg$ | 1104.9 | 9.9 | 466 | 2468 |
| $Ve$ | 14 | 3.2 | 28.6 | 161.2 |
| $Vp$ | 1118 | 13.1 | 490 | 2629 |
| $h_B^2$ (%) | 98.7 | 75.3 | 94.2 | 94 |

$Vg$; Genotypic variance, $Ve$; Environmental variance, $Vp$; Phenotypic variance, $h_B^2$; Broad sense heritability.

### 3.3. Univariate Models

#### 3.3.1. Univariate Parametric Stability Statistics (First-Year 2020)

The results of different univariate parametric stability statistics are given in Table 5. The stability parameter designed by Shukla ($\sigma^2$), Wricke's ecovalence ($Wi^2$), AMMI stability value (ASV), and AMMI stability index (ASI) are based on the concept that genotypes with the smallest stability value are the most stable ones. The stability values were worked out for rice genotypes over eight locations and are presented in Table 5. Based on $\sigma^2$, $Wi^2$, ASV, and ASI genotype, G1 (GSR-48) was found as the most stable genotype for plant height. Genotype G6 (IRRI-6) was found the most stable genotype for tillers per plant. Genotype G7 (Kissan basmati) was found most stable for grain yield per plant, and genotype G3 (GSR-112) was found as most stable for straw yield per plant. These genotypes are stable because their values are relatively close to zero.

#### 3.3.2. Univariate Parametric Stability Statistics (Second-Year 2021)

The univariate parametric stability statistics for 2021 found that a different trend for the stability of the same genotypes had changed from 2020. Using $\sigma^2$, $Wi^2$, ASV, and ASI stability indicators, genotype G4 (GSR-252) was identified as the most stable genotype for plant height. G3 (GSR-112) was also a stable genotype for plant height based on ASV and ASI values. Using $\sigma^2$ and $Wi^2$ values, genotype G3 (GSR-112) was found as the most stable genotype, and using ASV and ASI values, genotype G1 (GSR-48) was found to be the most stable genotype for tillers per plant. Genotype G4 (GSR-252) was found stable for grain yield per plant, as indicated by its lowest values for all studied stability statistics. Genotype G3 (GSR-112) was the most stable genotype based on $\sigma^2$ and $Wi^2$, while genotype G4 (GSR-252) was also identified as the stable genotype due to its lowest values for ASV and ASI.

### 3.4. Multivariate Models

#### 3.4.1. AMMI Analysis of Variance (First-Year 2020)

The AMMI model for yield and yield-related traits revealed significant variations ($p < 0.05$) for both the main (genotypes and locations) and interaction effects revealing the presence of considerable variability among the studied genotypes, locations, and their interactions (Supplementary Table S1). The maximum part of the total variance in the AMMI analysis was attributed to the locations factor, followed by genotypes and genotype by location interaction. In our study, locations explained the maximum (53%) of the total sum of squares for all traits, indicating that varied environmental conditions could cause most variations among genotype traits. Genotypes explained only 25% of the total sum of squares on average for traits, whereas the G × L interaction accounted for 20% of total variations.

The AMMI analysis generated two significant PCs from the G × L interaction. The PC1 and PC2 accounted for 80% of the variation for plant height, 73% for tillers per plant, 75% for grain yield per plant, and 84.5% for straw yield per plant, respectively (Supplementary Table S1). The extracted PCs are informative by elucidating information on the interaction effect; although, their degree decreases gradually from the first to the last PC.

**Table 5.** Parametric stability statistics for Plant height, Tillers per plant, Grain yield per plant, and Straw yield per plant of seven rice genotypes grown in eight different locations in Pakistan.

| Year 2020 | Plant Height | | | | Tillers per Plant | | | | Grain Yield per Plant | | | | Straw Yield per Plant | | | |
|---|---|---|---|---|---|---|---|---|---|---|---|---|---|---|---|---|
| Lines | $\sigma^2$ | $Wi^2$ | ASV | ASI | $\sigma^2$ | $Wi^2$ | ASV | ASI | $\sigma^2$ | $Wi^2$ | ASV | ASI | $\sigma^2$ | $Wi^2$ | ASV | ASI |
| G1 | −0.6 | 63.5 | 1.5 | 0.3 | 7.3 | 126.1 | 2.7 | 0.8 | 59.9 | 1277.4 | 2.3 | 0.5 | 399.5 | 8503.5 | 4.3 | 1.7 |
| G2 | 32.9 | 568.4 | 3.5 | 0.8 | 7.9 | 135.0 | 2.0 | 0.6 | 84.0 | 1637.9 | 6.5 | 1.4 | 672.7 | 12601.5 | 6.0 | 2.4 |
| G3 | 8.7 | 204.6 | 2.9 | 0.6 | 12.0 | 196.1 | 3.5 | 1.0 | 280.5 | 4586.2 | 11.4 | 2.6 | 269.2 | 6548.4 | 2.4 | 1.0 |
| G4 | 43.6 | 727.7 | 7.8 | 1.7 | 3.5 | 68.4 | 1.2 | 0.3 | 230.7 | 3839.2 | 10.6 | 2.4 | 352.8 | 7803.6 | 4.6 | 1.8 |
| G5 | 23.3 | 423.4 | 2.2 | 0.5 | 3.9 | 74.4 | 1.0 | 0.3 | 118.8 | 2159.7 | 4.6 | 1.0 | 1598.0 | 26481.4 | 8.4 | 3.4 |
| G6 | 33.0 | 569.8 | 6.6 | 1.5 | 0.6 | 25.5 | 0.5 | 0.1 | 74.5 | 1495.7 | 3.9 | 0.8 | 931.2 | 16478.9 | 6.2 | 2.5 |
| G7 | 30.6 | 533.2 | 6.3 | 1.4 | 1.2 | 34.2 | 1.1 | 0.3 | 32.7 | 869.0 | 0.5 | 0.1 | 1633.6 | 27014.9 | 8.6 | 3.5 |
| Year 2021 | Plant height | | | | Tillers Per Plant | | | | Grain Yield per Plant | | | | Straw Yield per Plant | | | |
| Lines | $\sigma^2$ | $Wi^2$ | ASV | ASI | $\sigma^2$ | $Wi^2$ | ASV | ASI | $\sigma^2$ | $Wi^2$ | ASV | ASI | $\sigma^2$ | $Wi^2$ | ASV | ASI |
| G1 | 22.5 | 484.5 | 4.1 | 0.6 | 2.1 | 52.3 | 0.6 | 0.1 | 74.1 | 1272.1 | 3.5 | 0.9 | 133.1 | 3218.3 | 2.5 | 0.7 |
| G2 | 95.7 | 1582.1 | 15.6 | 2.5 | 2.8 | 61.7 | 2.2 | 0.4 | 40.4 | 766.7 | 3.4 | 0.8 | 397.0 | 7176.7 | 7.7 | 2.1 |
| G3 | 29.3 | 585.5 | 3.0 | 0.4 | 0.6 | 29 | 1.7 | 0.3 | 70.0 | 1210.7 | 5.1 | 1.3 | −10.6 | 1061.5 | 1.8 | 0.5 |
| G4 | 8.3 | 270.1 | 3.0 | 0.4 | 7.9 | 138 | 3.4 | 0.7 | −1.2 | 141.2 | 0.9 | 0.2 | 87.3 | 2531.9 | 0.8 | 0.2 |
| G5 | 136.5 | 2194.3 | 18.3 | 2.9 | 5.2 | 97.8 | 2.3 | 0.4 | 55.2 | 988.2 | 2.7 | 0.7 | 765.7 | 12707.5 | 9.0 | 2.5 |
| G6 | 25.6 | 530.2 | 4.8 | 0.7 | 14.4 | 235 | 6.1 | 1.2 | 62.5 | 1098.0 | 5.4 | 1.4 | 1012.1 | 16403.0 | 12.4 | 3.4 |
| G7 | 21.7 | 472.5 | 5.8 | 0.9 | 13 | 215 | 5.1 | 1.0 | 70.6 | 1218.6 | 6.3 | 1.6 | 465.7 | 8208.4 | 4.7 | 1.3 |

$\sigma^2$: Shukla's stability variance; $Wi^2$: Wricke's Ecovalence for stability; ASV: AMMI Stability Value; ASI: AMMI Stability Index.

3.4.2. AMMI Analysis of Variance (Second-Year 2021)

We also conducted the AMMI analysis for the second year of the multi-location trials to reveal the effect of tested genotypes, locations, and their interaction with traits. Here, the AMMI model showed significant differences among tested genotypes, locations, and their interaction at ($p < 0.05$) probability for all the studied traits as analyzed in Supplementary Table S1. The greater contribution for the total sum of squares in AMMI analysis was caused by locations (66%), followed by genotype by location interaction effect (20%) and genotypes (11.8%). The maximum variation due to the interaction effect confirmed that tested genotypes responded significantly to the fluctuation in environmental conditions at locations. The proportion of PC1 and PC2 from the interaction effect explained 83% of the variation for plant height, 74.3% for tillers per plant, 72% for grain yield, and 75.5% for straw yield, respectively.

3.4.3. GGE Biplot Analysis (First-Year 2020)
'Mean vs. Stability' Analysis of GGE Biplot

The GSR lines' stability pattern across different locations was analyzed using the mean vs. stability pattern of the GGE biplot. It facilitates genotype evaluation based on mean performance and stability across various environments. The biplot graph is formed by the intersection of a vertical AEC abscissa and a horizontal AEC ordinate line. Each line has a single arrowhead that points towards a higher mean performance for the studied trait. In our investigation, the mean vs. stability analysis revealed 95.9% for plant height (Figure 1A), 75.66% for tillers per plant (Figure 1B), 75.63% for grain yield per plant (Figure 1C), and 88.34% variation for straw yield (Figure 1D), of G + G × E variation. Here, G5 (GSR-305) revealed maximum plant height in E2 (Kala Shah Kaku), E3 (Narowal), and E8 (Dokri); followed by two check cultivars, G6 (IRRI-6) and G7 (Kissan basmati) that showed maximum plant height in E1 (Pindi Bhattian), E4 (Swat), E5 (Islamabad), E6 (Dera Ismail Khan), and E7 (Muzaffargarh). G1 (GSR-48) and G3 (GSR-112) were the most stable GSR lines tested across different locations these lines recorded lower heights in all locations.

The maximum numbers of tillers per plant were recorded by a check cultivar G7 (Kissan basmati) followed by G4 (GSR-252) in E1 (Pindi Bhattian), E2 (Kala Shah Kaku), E3 (Narowal), E4 (Swat), and E5 (Islamabad). Only G3 (GSR-112) showed performance in E7 (Muzaffargarh). G2 (GSR-82), G6 (IRRI-6) and G5 (GSR-305) were the stable genotypes even though they produced fewer tillers and their performance is limited to the E8 (Dokri) location only (Figure 1B). G5 (GSR-305) was the most stable and high-performing genotype for grain yield per plant trait. The second-best high-performing genotype was G3 (GSR-112) in E8 (Dokri) and E7 (Muzaffargarh), even though it was not stable and the only genotype performing in E7 (Muzaffargarh). G1 (GSR-48) was the stable genotype after G5 (GSR-305) and, together with G5 (GSR-305), performed well in E1 (Pindi Bhattian), E2 (Kala Shah Kaku), E3 (Narowal), E4 (Swat), E5 (Islamabad), E6 (Dera Ismail Khan), and E8 (Dokri). Both check varieties and G4 (GSR-252) were not stable, and neither did they perform in any tested location for grain yield (Figure 1C). G5 (GSR-305) was also the best performing and most stable genotype for straw yield, followed by G3 (GSR-112) in E4 (Swat), E5 (Islamabad), and E8 (Dokri). G6 (IRRI-6) performed in E2 (Kala Shah Kaku), E3 (Narowal), E6 (Dera Ismail Khan), and E7 (Muzaffargarh). G1 (GSR-48), G4 (GSR-252), G7 (Kissan basmati), and G2 (GSR-82) showed some performance in E1 (Pindi Bhattian), but these were not stable genotypes for any tested location (Figure 1D).

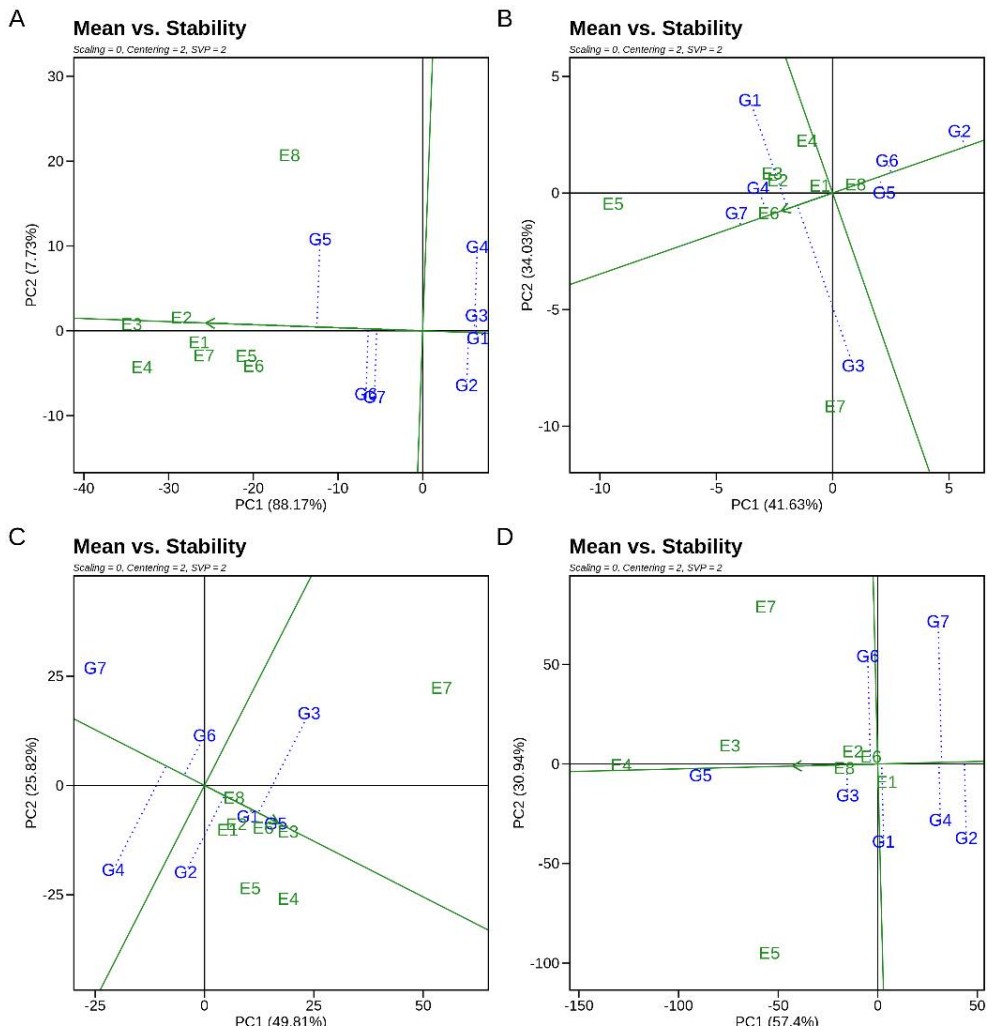

**Figure 1.** The GGE biplot 'Mean vs. stability' pattern of genotype × environment interaction of 5 GSR lines and 2 control lines grown under eight environments in the year 2020 for (**A**) plant height, (**B**) the number of tillers, (**C**) grain yield, and (**D**) straw yield. The biplots were created with cent− ering 0, SVP = 2, and scaling = 0 parameters.

'Which-Won-Where' GGE Biplot

Figure 2 represents the polygon view of the GGE biplot and it revealed the best performing genotypes for traits in a single group of locations. The G + G × E biplot ('which-won-where' pattern) explained 95.9%, 75.66%, 75.63%, and 88.34% variation for plant height, tillers per plant, grain yield per plant, and straw yield per plant, respectively (Figure 2). As explained by Oladosu et al. [27], the genotypes lying on the vertex of a polygon with no environmental indicator nearby are poorly performed genotypes, and the genotypes that are present on the vertex of a polygon where one or more environmental indicators are present are the best performing genotypes in the relevant environments. The genotypes lying inside a polygon are less responsive to any testing environment. All environmental indicators formed a single sector for plant height, and G5 (GSR-305) was the winning genotype in all testing environments. No other genotype won in any testing environment and thus defined poorly performing genotypes for plant height trait (Figure 2A). Eight environments were divided into four sectors for tillers per plant, with different genotypes winning in each sector. Sector one has environment E1 (Pindi Bhattian), E2 (Kala Shah Kaku), E3 (Narowal), and E4 (Swat); sector two has environment E5 (Islamabad) and E6 (Dera Ismail Khan); sector three has environment E7 (Muzaffargarh), and sector four has environment E8 (Dokri). G1 (GSR-48) was the winning genotype in sector one, G7 (Kissan

basmati) was the winning genotype in sector two, G3 (GSR-112) was the winning genotype in sector three, and G2 (GSR-82) was the winning genotype in sector four for tillers per plant (Figure 2B). The which-won-where GGE biplot of grain yield divided the eight locations into three sectors. Sector one has only environment E7 (Muzaffargarh) with G3 (GSR-112) the winning genotype; sector two has environment E1 (Pindi Bhattian), E2 (Kala Shah Kaku), E3 (Narowal), E4 (Swat), E6 (Islamabad), and E8 (Dokri) with G5 (GSR-305) the winning genotype in these environmental indicators; and sector three has environment E5 (Islamabad) with no winning genotype in it. G2 (GSR-82), G3 (GSR-112), and G7 (Kissan basmati) were poor-performing genotypes for grain yield (Figure 2C). For straw yield, which-won-where GGE biplot divided eight testing environments into three sectors. Sector one has environment E7 (Muzaffargarh) with no genotype winning in it; sector two has environment E2 (Kala Shah Kaku), E3 (Narowal), E4 (Swat), E5 (Islamabad), E6 (Dera Ismail Khan), and E8 (Dokri) with G5 (GSR-305) winning in all these testing environments; and sector three has environment E1 (Pindi Bhattian) with G2 (GSR-82) the winning genotype. Both check varieties were poorly performing genotypes for straw yield (Figure 2D).

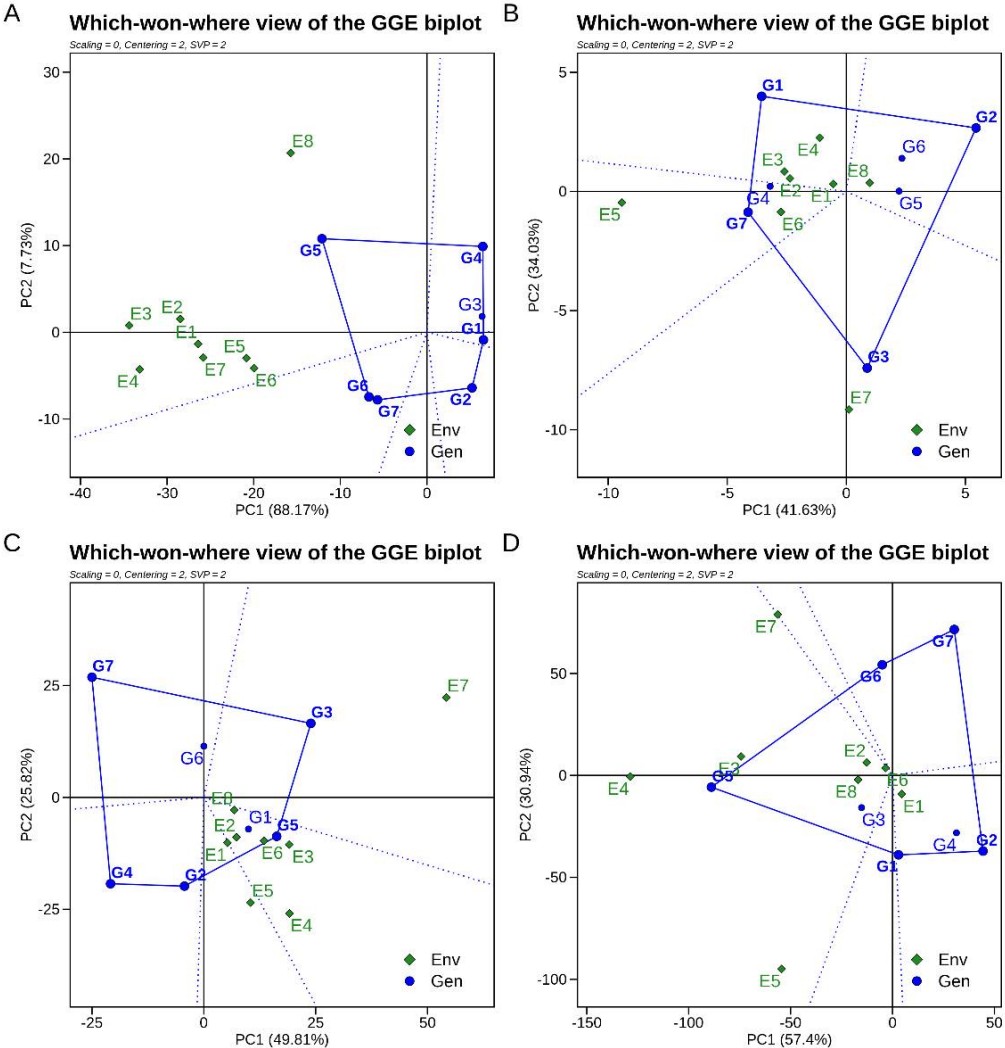

**Figure 2.** The GGE biplot polygon of the 'Which−won−where' pattern to identify the best cultivar in each location of 5 GSR lines and 2 control lines grown under eight environments in the year 2020 for (**A**) plant height, (**B**) the number of tillers, (**C**) grain yield, and (**D**) straw yield. The biplots were created with centering = 0, SVP = 2, and scaling = 0 parameters.

Locations and Genotypes Ranking: Best and Stable Location/Genotypes Evaluation

Figure 3 shows the GGE biplot 'Ranking environments' pattern to rank locations with respect to ideal environment or tester for genotypes. The genotypes are treated random and focus is placed on testing environments. E3 (Narowal) appeared to be as best locations for plant height (Figure 3A); E5 (Islamabad) for number of tillers (Figure 3B); and E4 (Swat) for both grain yield and straw yield (Figure 3C or Figure 3D). Testers were ranked based on their closeness to the concentric center. The rank of environments based on pattern of ranking environments GGE biplot for plant height is E3 > E2 ≈ E4 > E1 > E7 > E5 > E6 > E8; ranking for tillers per plant is E5 > E6 > E3 ≈ E2 > E1 > E4 > E8 > E7; ranking for grain yield is E4 > E3 > E5 ≈ E6 > E2 > E1 > E7 > E8; and ranking for straw yield is E4 > E3 > E7 ≈ E8 > E2 ≈ E5 > E6 > E1.

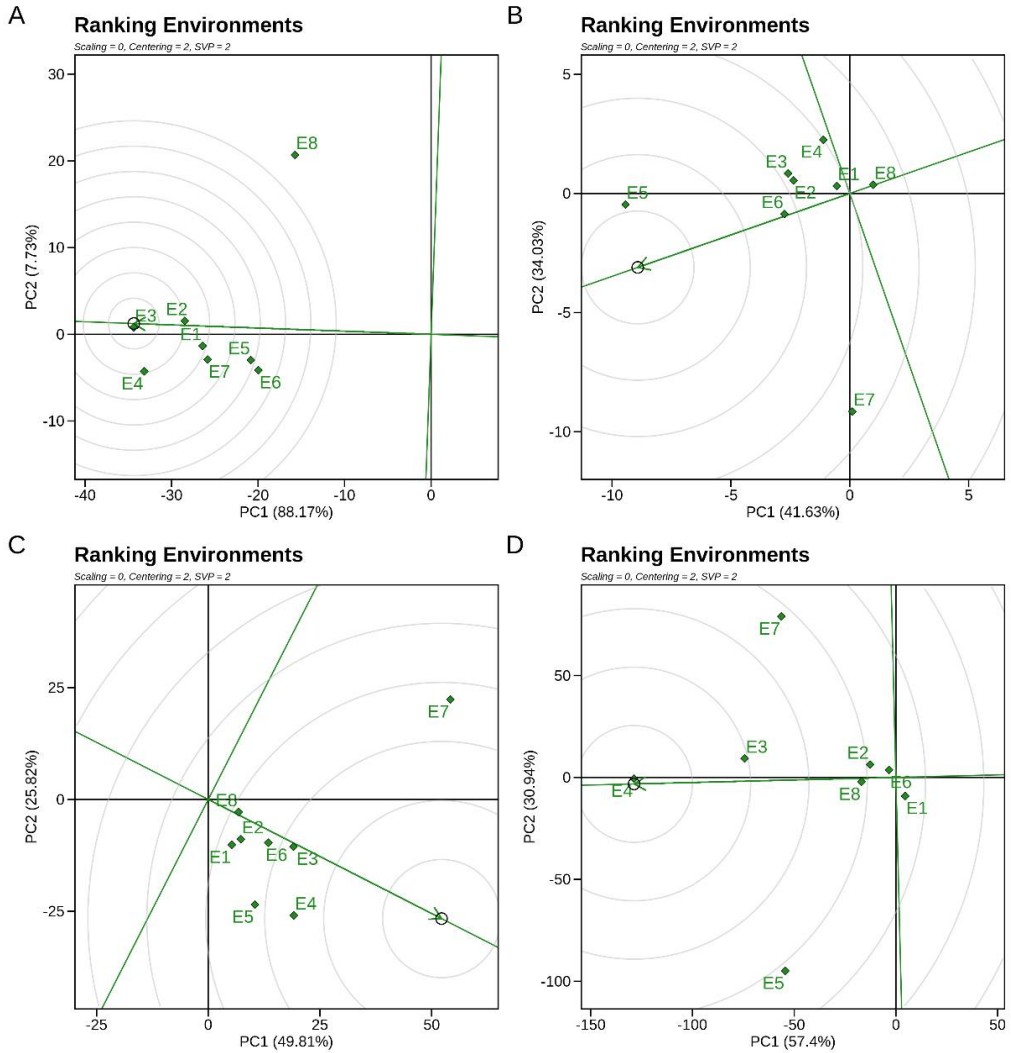

**Figure 3.** The GGE biplot 'Ranking environments' pattern to rank environments for the ideal env−ironment of 5 GSR lines and 2 control lines grown under eight environments in the year 2020 for (**A**) plant height, (**B**) the number of tillers, (**C**) grain yield, and (**D**) straw yield. The biplots were created with centering = 0, SVP = 2, and scaling = 0 parameters.

The GGE biplot of ranking genotypes concerning the ideal genotype revealed the unique genotype compared to the others evaluated (Figure 4). The blue arrowhead points toward the ideal genotype that performs best in all testing environments. Ideal entry is placed in the center of the concentric circle, followed by other circles. If no entry is located in the center, then the most closely located entry to the concentric circle is ideal. Environments are treated as random samples of testing environments, and the concentration points are

genotypes. G6 (IRRI-6) was the best genotype for plant height (Figure 4A) based on its nearness to the innermost circle. G7 (Kissan basmati) was the best genotype among others for tillers per plant (Figure 4B); for grain yield per plant, G5 (GSR-305) was the ideal genotype that was present within the innermost circle (Figure 4C). G5 (GSR-305) was also the best genotype for straw yield (Figure 4D). The genotypes ranking for plant height was G6 > G7 > G5 > G3 ≈ G1 ≈ G2 > G4; for tillers per plant G7 > G4 > G1 > G5 > G6 > G3 > G2; for grain yield G5 > G1 > G3 > G2 > G6 > G4 > G7; and for straw yield the ranks were G5 > G3 > G1 ≈ G6 > G4 > G2 ≈ G7.

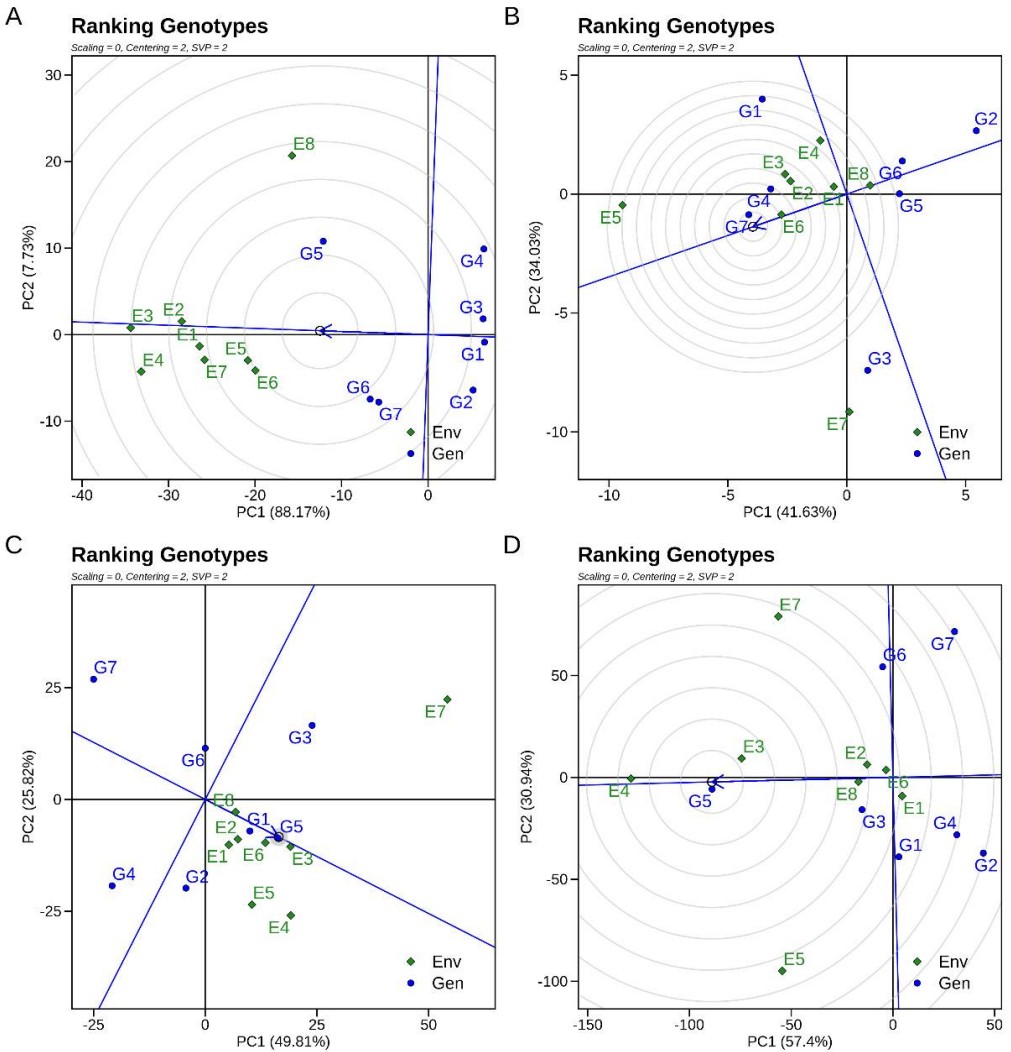

**Figure 4.** The GGE biplot 'Ranking genotypes' pattern to rank genotypes for the ideal genotype of 5 GSR lines and 2 control lines grown under eight environments in the year 2020 for (**A**) plant height, (**B**) the number of tillers, (**C**) grain yield, and (**D**) straw yield. The biplots were created with centering = 0, SVP = 2, and scaling = 0 parameters.

3.4.4. GGE Biplot Analysis (Second-Year 2021)
Mean vs. Stability' Analysis of GGE Biplot (First-Year 2021)

The genotype evaluation is based on the average performance and stability in various environments. The mean versus stability pattern of GGE biplots explained 90.23% for plant height, 69.74% for tillers per plant, 65.08% for grain yield, and 79.06% of the total variation for straw yield (Figure 5). Check variety G6 (IRRI-6) showed maximum plant height in E1 (Pindi Bhattian), and E3 (Narowal), followed by G5 (GSR-305) was high performing genotype in E2 (Kala Shah Kaku), E4 (Swat), E5 (Islamabad), E6 (Dera Ismail Khan), E7 (Muzaffargarh), and E8 (Dokri). G7 (Kissan basmati) performed better in E1 (Pindi Bhattian)

and E3 (Narowal). G3 (GSR-112) and G4 (GSR-252) were the stable genotypes even though with less performance (Figure 5A). G4 (GSR-252) produced the maximum number of tillers per plant in E1 (Pindi Bhattian), E2 (Kala Shah Kaku), and E3 (Narowal). After that, G5 (GSR-305) and check variety G6 (IRRI-6) showed good performance in E4 (Swat), E5 (Islamabad), E6 (Dera Ismail Khan), E7 (Muzaffargarh), and E8 (Dokri). G1 (GSR-48), G2 (GSR-82), and G3 (GSR-112) were the stable genotypes with fewer tillers. G5 (GSR-305) was also a stable genotype with good performance (Figure 5B). G3 (GSR-112) followed by G1 (GSR-48) and G5 (GSR-305) were high grain yielding genotypes in G7 (Muzaffargarh). G7 (Kissan basmati) was the stable genotype for grain yield in E2 (Kala Shah Kaku), E3 (Narowal), and E6 (Dera Ismail Khan). G3 (GSR-112), G2 (GSR-82), and G4 (GSR-252) showed performance in E1 (Pindi Bhattian), E4 (Swat), and E5 (Islamabad). At the same time, G5 (GSR-305) and G1 (GSR-48) were performing genotypes in E7 (Muzaffargarh) (Figure 5C). G5 (GSR-305) was high performing genotype for straw yield in E2 (Kala Shah Kaku), E3 (Narowal), and E7 (Muzaffargarh). In environments E1 (Pindi Bhattian), E5 (Islamabad), E6 (Dera Ismail Khan), and E8 (Dokri), the only performing genotype was check cultivar G6 (IRRI-6). G1 (GSR-48), G3 (GSR-112), and G4 (GSR-252) were the stable genotypes. In E4 (Swat) some performance was shown by G1 (GSR-48), G2 (GSR-82), G3 (GSR-112), G4 (GSR-252), and G7 (Kissan basmati) (Figure 5D).

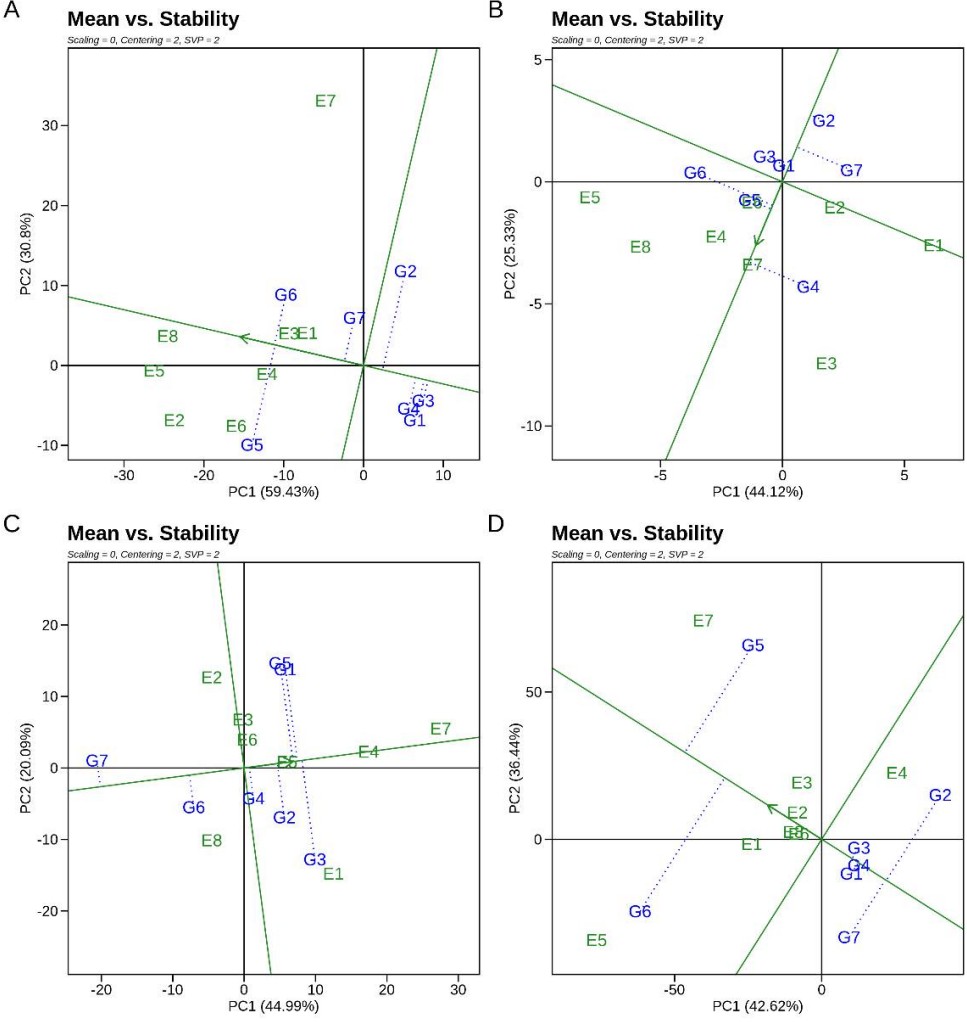

**Figure 5.** The GGE biplot 'Mean versus stability' pattern of genotype × environment interaction of 5 GSR lines and 2 control lines grown under eight environments in the year 2021 for (**A**) plant height, (**B**) the number of tillers, (**C**) grain yield, and (**D**) straw yield. The biplots were created with centering = 0, SVP = 2, and scaling = 0 parameters.

'Which-Won-Where' GGE Biplot

The PC1 and PC2 scores of the constructed GGE biplot of 'which-won-where' for the year 2021 explained 90.23%, 69.74%, 65.08%, and 79.06% of total variations for plant height, tillers per plant, grain yield per plant, and straw yield per plant, respectively (Figure 6). The genotypes positioned at the corners of the polygons for the studied traits were considered elite in that location. The genotypes placed at vertexes with no tester are regarded as poor genotypes. This GGE biplot divided eight testing environments into three sectors for plant height traits. Sector one has one environment E7 (Muzaffargarh) with G2 (GSR-82) the winning genotype in it; sector two has E1 (Pindi Bhattian) and E3 (Narowal) with check variety G6 (IRRI-6) winning in it; and sector three has E2 (Kala Shah Kaku), E4 (Swat), E5 (Islamabad), E6 (Dera Ismail Khan), and E8 (Dokri) with G5 (GSR-305) as the winning genotype (Figure 6A). For tillers per plant, testing environments formed two sectors. Sector one has E4 (Swat), E5 (Islamabad), E6 (Dera Ismail Khan), and E8 (Dokri) with check variety G6 (IRRI-6) as the winning genotype in these testing environments; and sector two has E1 (Pindi Bhattian), E2 (Kala Shah Kaku), E3 (Narowal), and E7 (Muzaffargarh) with G4 (GSR-252) as a winning genotype for these environmental indicators (Figure 6B). The which-won-where GGE biplot for grain yield divided the eight testers into four sectors. G1 (GSR-48) and G5 (GSR-305) were winning genotypes in sector one that has environment E2 (Kala Shah Kaku), E3 (Narowal), and E6 (Dera Ismail Khan). Sector two has environment E7 (Muzaffargarh) with no winning genotype. G3 (GSR-112) was the winning genotype in sector three that has environment E1 (Pindi Bhattian), E4 (Swat), and E5 (Islamabad). Sector four has an environment with no genotype winning in it. Both check varieties were poorly performing for grain yield (Figure 6C). The which-won-where pattern of straw yield separated eight testers into three sectors. Sector one has environment E2 (Kala Shah Kaku), E3 (Narowal), and E7 (Muzaffargarh) with G5 (GSR-305) the winning genotype in these environments; sector two has environment E4 (Swat) with G2 (GSR-82) as the winning genotype; and sector three has environment E1 (Pindi Bhattian), E5 (Islamabad), E6 (Dera Ismail Khan), and E8 (Dokri) with check variety G6 (IRRI-6) as the winning genotype in these testers. G7 (Kissan Basmati) was regarded poorly performing genotype for straw yield (Figure 6D).

Locations and Genotypes Ranking: Best and Stable Location/Genotypes Evaluation

Ranking location pattern of GGE biplots reveals ideal testing environments for all entries. The green arrow points towards the ideal environment, which is placed in the inner most circle. Genotypes are treated as random samples of entries and focus is placed on testers. In our investigation for the second year E8 (Dokri) was found ideal location for genotypes plant height (Figure 7A) and E7 (Muzaffargarh) for tillers per plant, grain yield, and straw yield (Figure 7B–D). The rank of environments based on pattern of ranking environments GGE biplot for plant height is E8 > E5 > E2 > E4 ≈ E6 > E3 > E1 > E7; ranking for tillers per plant is E7 > E3 ≈ E4 > E8 > E6 > E2 ≈ E5 > E1; ranking for grain yield is E7 > E4 > E5 > E1 > E6 > E3 > E2 > E8; and ranking for straw yield is E7 > E1 ≈ E3 > E2 > E8 > E6 ≈ E5 > E4.

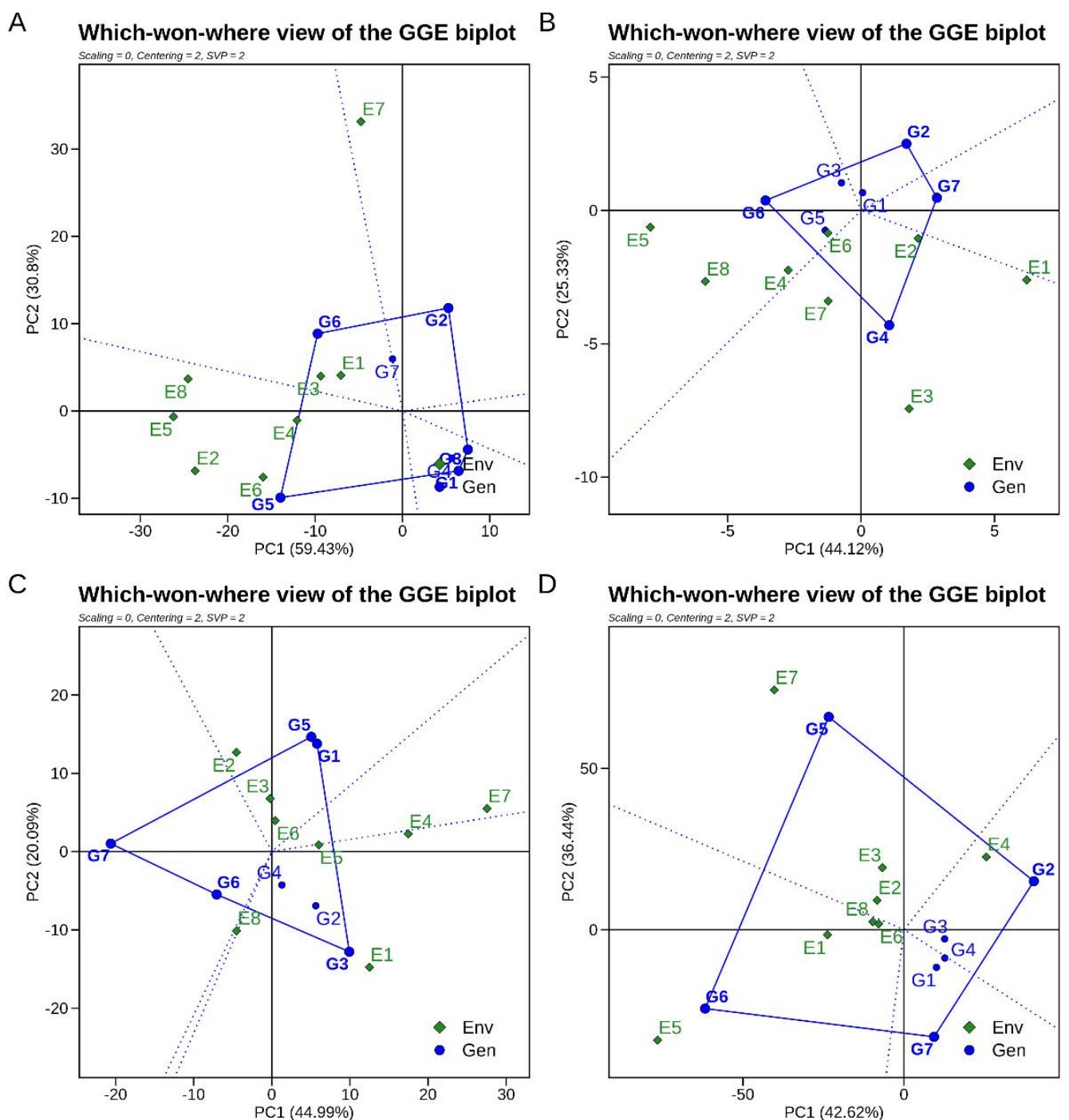

**Figure 6.** The GGE biplot polygon of the 'Which-won-where' pattern to identify the best cultivar in each location of 5 GSR lines and 2 control lines grown under eight environments year in 2021 for (**A**) plant height, (**B**) the number of tillers, (**C**) grain yield, and (**D**) straw yield. The biplots were created with centering = 0, SVP = 2, and scaling = 0 parameters.

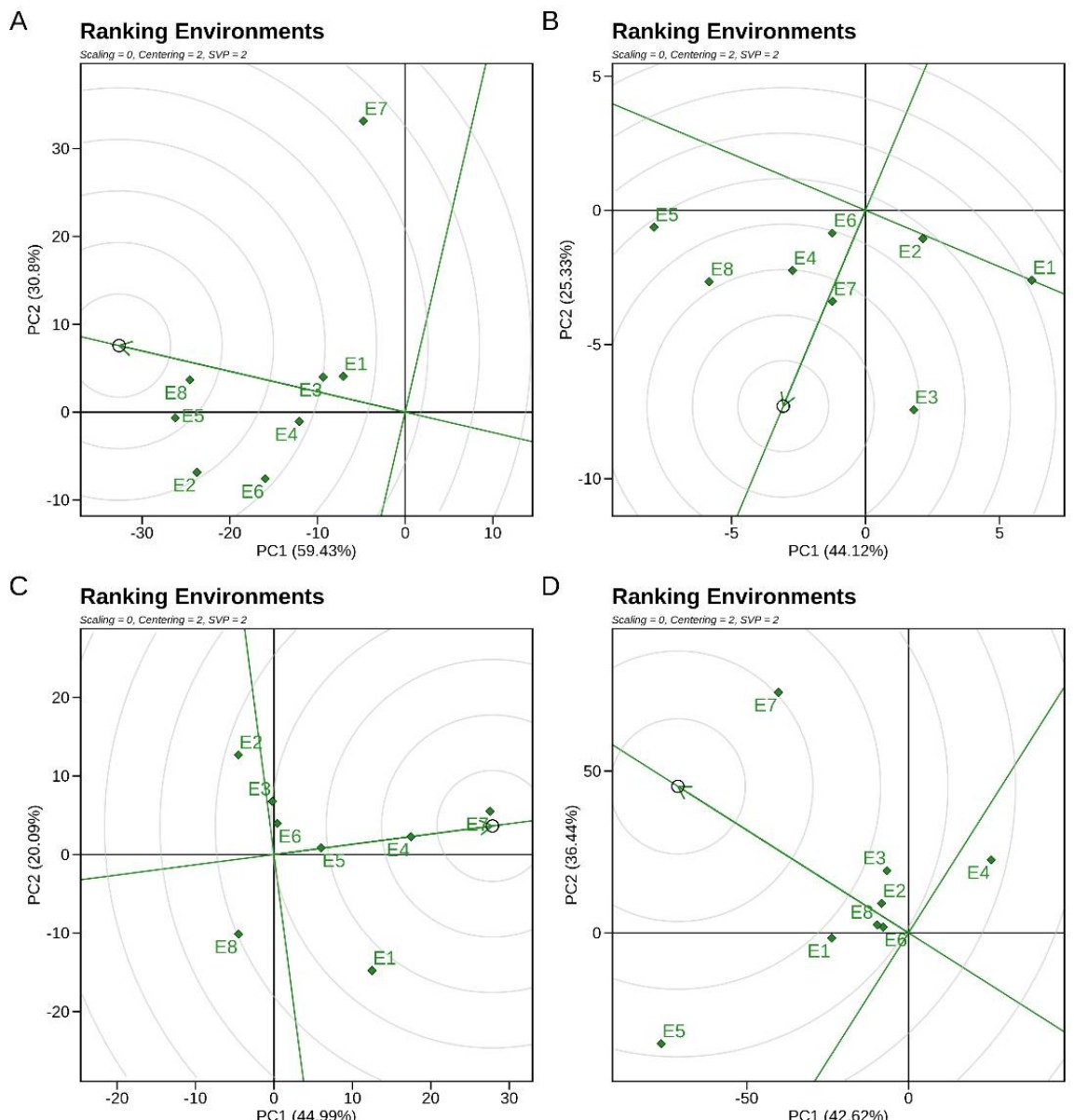

**Figure 7.** The GGE biplot 'Ranking environments' pattern to rank environments for an ideal environment of 5 GSR lines and 2 control lines grown under eight environments in the year 2021 for (**A**) plant height, (**B**) the number of tillers, (**C**) grain yield, and (**D**) straw yield. The biplots were created with centering = 0, SVP = 2, and scaling = 0 parameters.

Using the genotype ranking GGE biplot (Figure 8) we can identify the best entry in comparison to other entries tested in all testers. GGE biplot noted G6 (IRRI-6) as high performing genotypes for plant height (Figure 8A); G5 (GSR-305) for tillers per plant (Figure 8B); G5 (GSR-305) for grain yield per plant (Figure 8C); and again G5 (GSR-305) for straw yield per plant (Figure 8D). The genotypes ranking for plant height was G6 > G7 > G5 > G4 > G1 > G2 ≈ G3; for tillers per plant G5 > G4 > G1 > G6 ≈ G3 > G7 > G2; for grain yield G5 > G1 > G3 > G2 > G6 > G4 > G7; and for straw yield the ranks were G5 > G3 > G1 ≈ G6 > G4 > G2 ≈ G7.

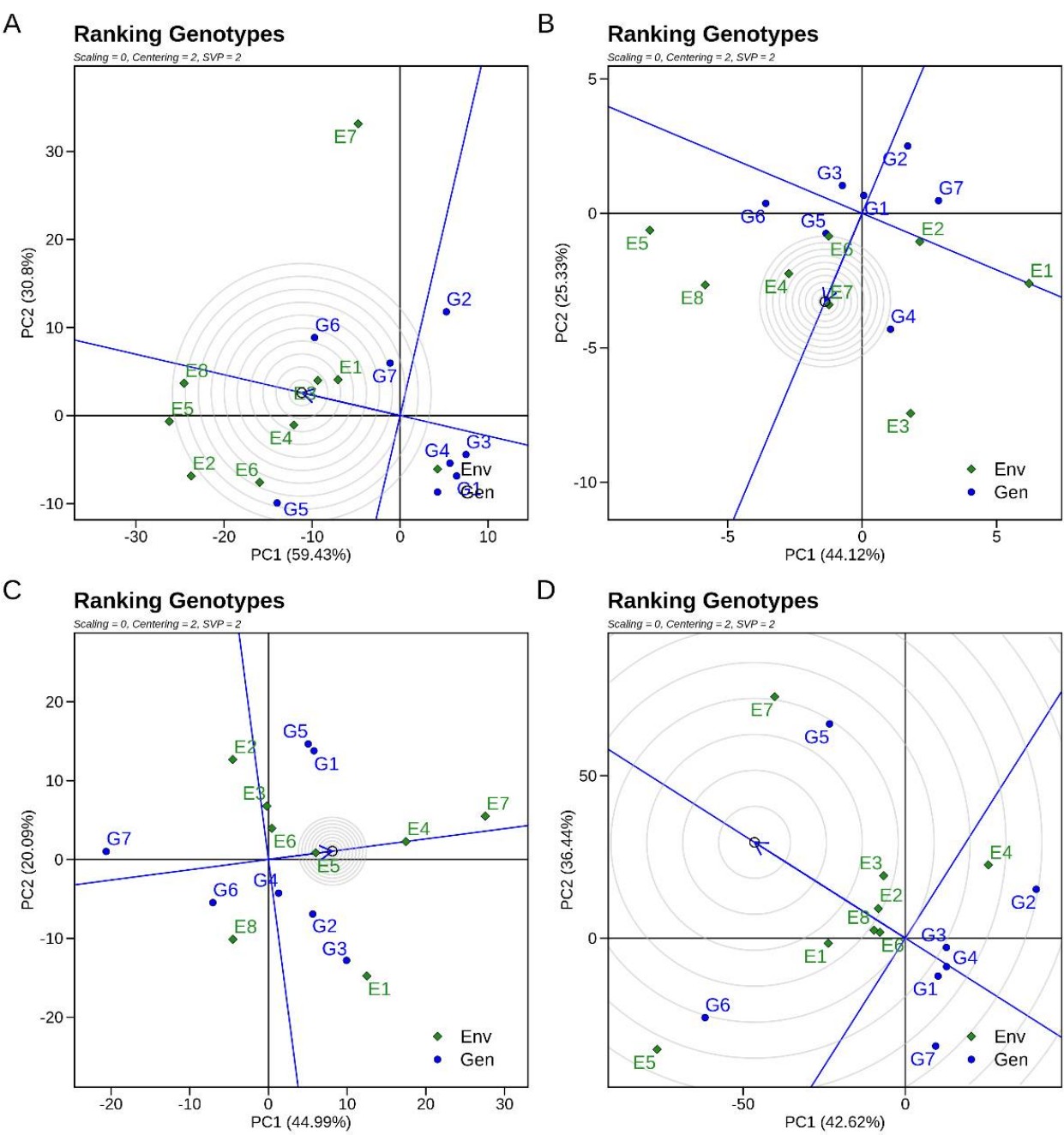

**Figure 8.** The GGE biplot 'Ranking genotypes' pattern to rank genotypes for the ideal genotype of 5 GSR lines and 2 control lines grown under eight environments in the year 2021 for (**A**) plant height, (**B**) the number of tillers, (**C**) grain yield, and (**D**) straw yield. The biplots were created with centering = 0, SVP = 2, and scaling = 0 parameters.

## 4. Discussion

The Green Super Rice in Pakistan (GSRP) project is one of the research components of megaprojects on "Productivity Enhancement of Rice" in Pakistan, where the task is to rapidly increase rice grain yield from 10 to 20 t/ha. The pedigree of the newly introduced GSR advanced lines is the mixture of more than 250 different potential rice varieties and hybrids adapted to challenging growing conditions. The GSR breeding project has been successful in attaining the most satisfactory traits, including strong and erect plant architecture, early maturity, maximum tillering, long and dense panicle, and disease/insect pest resistance (Figure 9).

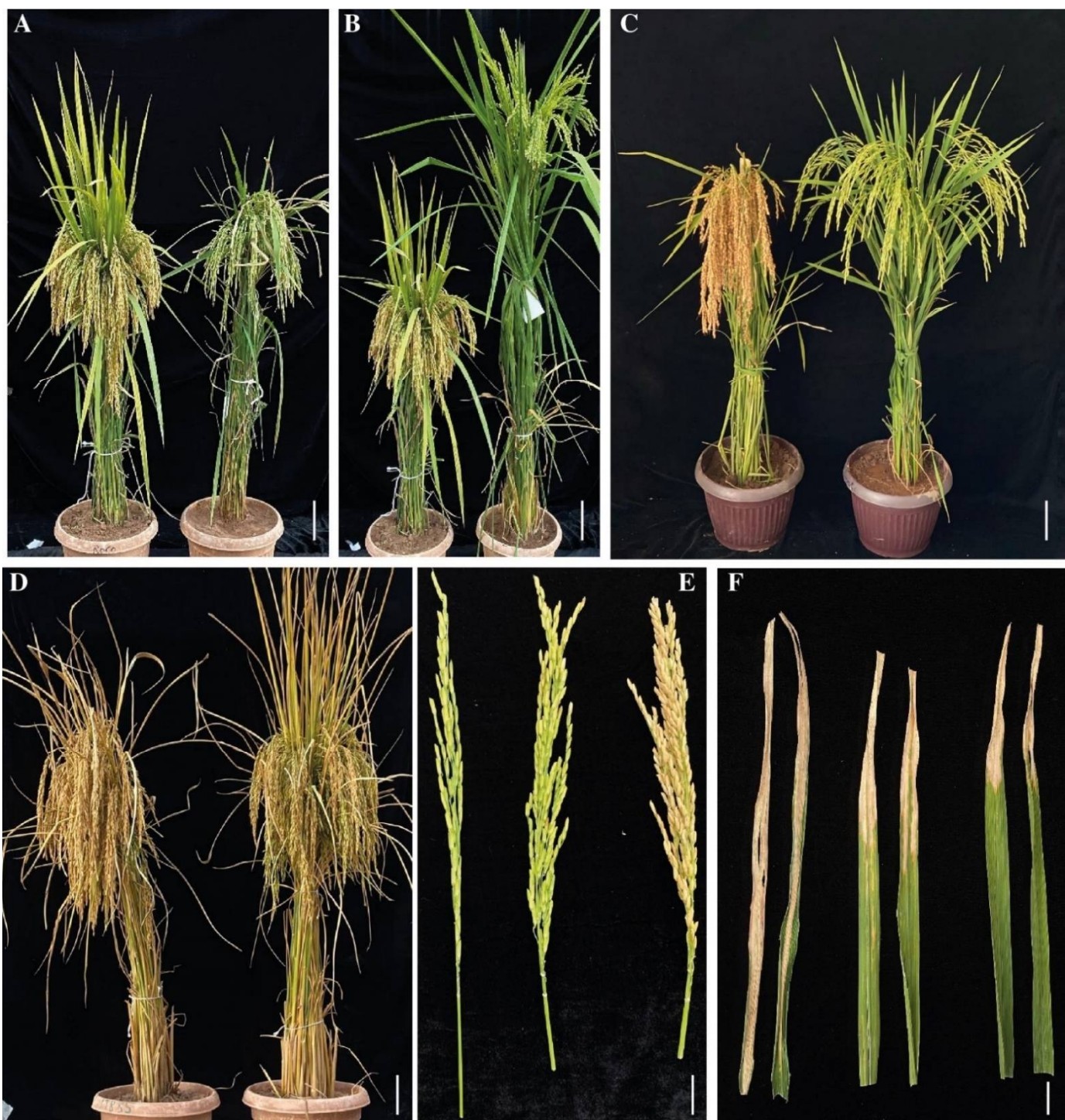

**Figure 9.** Salient features of GSR traits over BASMATI rice: (**A**) Architecture, left-side plant is GSR and right-side plant is BASMATI, scale = 10 cm; (**B**) Stature, short BASMATI plant vs. long GSR plant, scale = 10 cm; (**C**) Maturity, early maturing GSR plant vs. late maturing BAS-MATI plant, scale = 10 cm; (**D**) Tillering, left-side plant is BASMATI and right-side plant is GSR, scale = 10 cm; (**E**) Panicle density, left-side plant is BASMATI, center and right-side plant are GSR, scale = 1 cm; (**F**) Bacterial blight disease, left-side plant is BASMATI, center and right-side plant are GSR, scale = 1 cm.

Univariate stability parameters: AMMI stability value (ASV), AMMI stability index (ASI), Shukla ($\sigma^2$), Wricke's ecovalence ($W_i{}^2$), multivariate stability parameters; AMMI-

model and GGE biplots, were determined to find out the stable GSR line. GSR 48 was identified as the most stable genotype as a result of univariate stability analysis, while multivariate analyses have identified GSR 305 and GSR 252 as the most stable genotypes. Haider et al. [28] evaluated 18 rice varieties for yield and stability in Pakistan using the data from Rice Research Institute, Kala Shah Kaku for two years over nine different environments [28]. Our results demonstrated that the tested genotypes across different locations for two consecutive years are highly vulnerable to climatic zones and environmental factors [29,30]. Such variances could be due to the difference in topography and climatic conditions across different locations and years where the experimentations were conducted [31]. The breeding protocol must quantify genotype, environment, years, and their interaction factors to obtain successful breeding results of yield and related traits in rice [32]. The present findings of significant sources of variation have been previously noted in rice [33,34] and other cereal crops [35,36].

Stability analysis for multi-location data has been evaluated in both univariate and multivariate statistics [37]. Among the multivariate methods, the additive main effects and AMMI analysis are widely used for G × E interactions. The AMMI model combines ANOVA and G × E interactions to identify the genotypes and environmental variables [23,26]. The relative contributions of the total sum of squares of location, genotype, and GL interactions in the AMMI model of two-year data for grain yield per plant showed a similar pattern in the previous rice stability analysis [31,38]. Significant interactions between locations and tested genotypes in plant height and tillers per plant, as a high portion of the first two interaction principle components (IPCA1 and IPCA2), have been reported [32].

In our study, the univariate stability analysis screened out highly stable (GSR 112 and GSR 252) GSR lines for most of the studied traits. The GGE biplot analysis showed that IIRI-6 was the most stable genotype for plant height. GSR-305 and Kissan basmati were the most stable genotypes for tillers per plant. GSR 305 was closed to the biplot origin, depicting less response than the vertex genotypes. Moreover, it also reveals low environmental interaction in terms of grain and straw yield per plant. On the contrary, the other genotypes were farther from the biplot origin and demonstrated higher vulnerability towards environmental factors that affect their stability. Based on the adaptation pattern, Narowal and Dokri were found to be the most dynamic locations for genotypes plant height, Muzaffargarh and the NARC for tillers per plant, and Swat and Muzaffargarh for grain and straw yield per plant in 2020 and 2021, respectively. However, the tested genotypes showed different yields concerning their locations for the yield traits. Similar observations of the biplot model for multi-location studies using rice genotypes were also concluded earlier [39,40]. However, high-performing GSR lines for yield traits with less stability across locations can be stabilized following the backcross approach [41] with the most stable GSR line.

## 5. Conclusions

In our study, multi-location adoptability trials were aimed to predict the most promising rice genotypes across multi-environmental conditions in Pakistan. In this regard, several univariate and multivariate parametric stability models were analyzed to determine the stability performances of genotypes across environments. This study revealed three consistently stable GSR lines with minimum stability values in univariate stability statistics: GSR 305, GSR 252, and GSR 112. It is noted that GSR 48 showed the maximum stability when compared to all other lines in the univariate model across the two years for grain yield and related traits data. Furthermore, it is also concluded that multivariate parametric stability models (AMMI analysis of variance and GGE biplot) are great components to select the most suitable and stable GSR lines for specific as well as diverse environments. In this study, the combined ANOVA of the AMMI model showed that genotypes, locations, G × L interactions, and AMMI components (PCs one and two) were found significant. Therefore, yield and significant PCs were taken into account simultaneously to define the effect of GL

interactions and, then, to predict the most stable GSR line. Resultantly, AMMI and GGE biplot analysis classified GSR 305 and GSR 252 as the most stable genotypes across eight tested locations. Moreover, Swat, Narowal, and Muzaffargarh tend to be the best locations to commercialize GSR lines in Pakistan.

**Supplementary Materials:** The following supporting information can be downloaded at: https://www.mdpi.com/article/10.3390/agronomy12051157/s1, Supplementary Table S1: Combined analysis of variances and AMMI stability model.

**Author Contributions:** M.R.K. and I.U.Z. initiated the concept. M.H., M.K.N., U.A., M.U., A.L. and A.R. contributed in conducted research and data collection. M.R.K. and G.M.A. acquired the funding and other resources. I.U.Z. and N.Z. wrote the paper. M.R.K. and M.U. helped with manuscript revision. M.R.K. supervised the experiment. All authors have read and agreed to the published version of the manuscript.

**Funding:** This research was funded by the PSDP project "Productivity Enhancement of rice subcomponent Green Super Rice in Pakistan" at the National Institute for Genomics and Advanced Biotechnology, National Agriculture Research Centre Islamabad.

**Data Availability Statement:** All the data are presented in the main text and supplementary file.

**Acknowledgments:** We thank Zhikang Li, Jianlong Xu, and Shahzad Amir Naveed from the Institute of Crop Sciences, Chinese Academy of Agricultural Sciences (CAAS), who provided germplasm and valuable suggestions about Green Super Rice breeding in Pakistan.

**Conflicts of Interest:** The authors declare no conflict of interest.

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
