# Peer review of "Estimation of Genetic Variances and Stability Components of Yield-Related Traits of Green Super Rice at Multi-Environmental Conditions in Pakistan"

_agronomy, doi:10.3390/agronomy12051157_

Round 1

Reviewer 1 Report

The manuscript has been significantly improved compared to the previous version. However, still one of the basic disadvantages is the descriptive nature of the paper. There is no assessment of the relationship between the studied traits and the values of the calculated parameters, which would be interesting for other researchers and breeders of this species. Certainly, finding a correlation between the stability of the yield and other yield-generating traits would contribute to the genetic progress of this species. Also the conclusions (summary) are very general, making it impossible to state what the authors bring to the progress of rice breeding.

Reviewer 2 Report

please see my attachment, for most of the points from previous I could not identify that they were considered

Author Response

This manuscript is a resubmission of an earlier submission. The following is a list of the peer review reports and author responses from that submission.

Round 1

Reviewer 1 Report

One of the goals of the Green Super Rice project is to produce varieties that are stable and of good quality. The manuscript serves these purposes by assessing potential genotypes that can be considered a GSR. It is surprising that only 5 such genotypes were considered (plus two control variants). The assessment of the used genetic parameters or the AMMI model with a relatively small number of genotypes is not very reliable and will not bring anything new to research on the breeding of this species. Besides, the work has a strong descriptive character. There is evaluated for relationships between the response of genotypes and the prevailing conditions in research locations or weather conditions. It would be interesting to study how the tested varieties behave over two growing seasons and not separately for the season. Therefore, in many studies in this field, we consider combinations of years and locations as the environment to more reliably assess stability and other interaction GE parameters.

Minor comments

- The lack of presentation of the study ANOVA model difficult of  the assessment of its correctness

- The use of abbreviations of varieties and locations in theFigure that are not used in the text makes it impossible to follow the results with their description.

Reviewer 2 Report

Dear authors,

the manuscript has an above-average quality. Nevertheless, I have general concens about the amount and quality of the data that can make a publication of the data impossible. Beside this, there are several ways to enhence the statistical analysis.

Reviewer 3 Report

The manuscript entitled “Estimation of genetic variances, heterotic effects, AMMI model and its components for yield stability of Green Super Rice in multi-environmental conditions of Pakistan.” authored by Zaid et al. could be interesting for the adaptation of the GSR variety for the diversified agricultural system of Pakistan. However, I am sorry to decline to accept the manuscript for publication in Agronomy for the following reason:

  1. The plot size is extremely small for reliable yield to analyze yield stability; therefore, the manuscript uses yield per plant. Even the authors don’t mention any grain yield and straw yield unit. However, there are no commonly accepted guidelines for assessing yield stability. But such a small-scale field experiment in different locations can question the quantification of stability, especially the yield, which is the key component of the yield stability experiment.
  2. The discussion section is not well written.